# Online Dynamic Programming

**Holakou Rahmanian**
Department of Computer Science
University of California Santa Cruz
Santa Cruz, CA 95060
holakou@ucsc.edu

**Manfred K. Warmuth**
Department of Computer Science
University of California Santa Cruz
Santa Cruz, CA 95060
manfred@ucsc.edu

## Abstract

We consider the problem of repeatedly solving a variant of the same dynamic programming problem in successive trials. An instance of the type of problems we consider is to find a good binary search tree in a changing environment. At the beginning of each trial, the learner probabilistically chooses a tree with the $n$ keys at the internal nodes and the $n + 1$ gaps between keys at the leaves. The learner is then told the frequencies of the keys and gaps and is charged by the average search cost for the chosen tree. The problem is online because the frequencies can change between trials. The goal is to develop algorithms with the property that their total average search cost (loss) in all trials is close to the total loss of the best tree chosen in hindsight for all trials. The challenge, of course, is that the algorithm has to deal with exponential number of trees. We develop a general methodology for tackling such problems for a wide class of dynamic programming algorithms. Our framework allows us to extend online learning algorithms like *Hedge* [16] and *Component Hedge* [25] to a significantly wider class of combinatorial objects than was possible before.

## 1   Introduction

Consider the following *online learning* problem. In each trial, the algorithm plays with a Binary Search Tree (BST) for a given set of $n$ keys. Then the adversary reveals a set of probabilities for the $n$ keys and their $n + 1$ gaps, and the algorithm incurs a linear loss of *average search cost*. The goal is to predict with a sequence of BSTs minimizing *regret* which is the difference between the total loss of the algorithm and the total loss of the single best BST chosen in hindsight.

A natural approach to solve this problem is to keep track of a distribution on all possible BSTs during the trials (e.g. by running the *Hedge* algorithm [16] with one weight per BST). However, this seems impractical since it requires maintaining a weight vector of exponential size. Here we focus on combinatorial objects that are comprised of $n$ components where the number of objects is typically exponential in $n$. For a BST the components are the depth values of the keys and the gaps in the tree. This line of work requires that the loss of an object is linear in the components (see e.g. [35]). In our BST examples the loss is simply the dot product between the components and the frequencies.

There has been much work on developing efficient algorithms for learning objects that are composed of components when the loss is linear in the components. These algorithms get away with keeping one weight per component instead of one weight per object. Previous work includes learning $k$-sets [36], permutations [19, 37, 2] and paths in a DAG [35, 26, 18, 11, 5]. There are also general tools for learning such combinatorial objects with linear losses. The *Follow the Perturbed Leader (FPL)* [22] is a simple algorithm that adds random perturbations to the cumulative loss of each component, and then predicts with the combinatorial object that has the minimum perturbed loss. The *Component Hedge (CH)* algorithm [25] (and its extensions [34, 33, 17]) constitutes another generic approach. Each object is typically represented as a bit vector over the set of components where the 1-bits

indicate the components appearing in the object. The algorithm maintains a mixture of the weight vectors representing all objects. The weight space of CH is thus the convex hull of the weight vectors representing the objects. This convex hull is a polytope of dimension $n$ with the objects as corners. For the efficiency of CH it is typically required that this polytope has a small number of facets (polynomial in $n$). The CH algorithm predicts with a random corner of the polytope whose expectation equals the maintained mixture vector in the polytope.

Unfortunately the results of CH and its current extensions cannot be directly applied to problems like BST. This is because the BST polytope discussed above does not have a characterization with polynomially many facets. There is an alternate polytope for BSTs with a polynomial number of facets (called the *associahedron* [29]) but the average search cost is not linear in the components used for this polytope. We close this gap by exploiting the dynamic programming algorithm which solves the BST optimization problem. This gives us a polytope with a polynomial number of facets while the loss is linear in the natural components of the BST problem.

**Contributions**     We propose a general method for learning combinatorial objects whose optimization problem can be solved efficiently via an algorithm belonging to a wide class of dynamic programming algorithms. Examples include BST (see Section 4.1), Matrix-Chain Multiplication, Knapsack, Rod Cutting, and Weighted Interval Scheduling (see Appendix A). Using the underlying graph of subproblems induced by the dynamic programming algorithm for these problems, we define a representation of the combinatorial objects by encoding them as a specific type of subgraphs called *k-multipaths*. These subgraphs encode each object as a series of successive decisions (i.e. the components) over which the loss is linear. Also the associated polytope has a polynomial number of facets. These properties allow us to apply the standard Hedge [16, 28] and Component Hedge algorithms [25].

**Paper Outline**     In Section 2 we start with online learning of paths which are the simplest type of subgraphs we consider. This section briefly describes the main two existing algorithms for the path problem: (1) An efficient implementation of Hedge using *path kernels* and (2) Component Hedge. Section 3 introduces a much richer class of subgraphs, called *k-multipaths*, and generalizes the algorithms. In Section 4, we define a class of combinatorial objects recognized by dynamic programming algorithms. Then we prove that minimizing a specific dynamic programming problem from this class over trials reduces to online learning of $k$-multipaths. The online learning for BSTs uses $k$-multipaths for $k = 2$ (Section 4.1). A large number of additional examples are discussed in Appendix A. Finally, Section 5 concludes with comparison to other algorithms and future work and discusses how our method is generalized for arbitrary "min-sum" dynamic programming problems.

## 2   Background

Perhaps the simplest algorithms in online learning are the "experts algorithms" like the Randomized Weighted Majority [28] or the Hedge algorithm [16]. They keep track of a probability vector over all experts. The weight/probability $w_i$ of expert $i$ is proportional to $\exp(-\eta\, L(i))$, where $L(i)$ is the cumulative loss of expert $i$ until the current trial and $\eta$ is a non-negative learning rate. In this paper we use exponentially many combinatorial objects (composed of components) as the set of experts. When Hedge is applied to such combinatorial objects, we call it *Expanded Hedge (EH)* because it is applied to a combinatorially "expanded domain". As we shall see, if the loss is linear over components (and thus the exponential weight of an object becomes a product over components), then this often can be exploited for obtaining an efficient implementations of EH.

**Learning Paths**     The online shortest path has been explored both in full information setting [35, 25] and various bandit settings [18, 4, 5, 12]. Concretely the problem in the full information setting is as follows. We are given a directed acyclic graph (DAG) $\mathcal{G} = (V, E)$ with a designated source node $s \in V$ and sink node $t \in V$. In each trial, the algorithm predicts with a path from $s$ to $t$. Then for each edge $e \in E$, the adversary reveals a loss $\ell_e \in [0, 1]$. The loss of the algorithm is given by the sum of the losses of the edges along the predicted path. The goal is to minimize the regret which is the difference between the total loss of the algorithm and that of the single best path chosen in hindsight.

**Expanded Hedge on Paths** Takimoto and Warmuth [35] found an efficient implementation of EH by exploiting the additivity of the loss over the edges of a path. In this case the weight $w_\pi$ of a path $\pi$ is proportional to $\prod_{e \in \pi} \exp(-\eta L_e)$, where $L_e$ is the cumulative loss of edge $e$. The algorithm maintains one weight $w_e$ per edge such that the total weight of all edges leaving any non-sink node sums to 1. This implies that $w_\pi = \prod_{e \in \pi} w_e$ and sampling a path is easy. At the end of the current trial, each edge $e$ receives additional loss $\ell_e$, and the updated path weights have the form $w_\pi^{\text{new}} = \frac{1}{Z} \prod_{e \in \pi} w_e \exp(-\eta \ell_e)$, where $Z$ is a normalization. Now a certain efficient procedure called *weight pushing* [31] is applied. It finds new edge weights $w_e^{\text{new}}$ s.t. the total outflow out of each node is one and the updated weights are again in "product form", i.e. $w_\pi^{\text{new}} = \prod_{e \in \pi} w_e^{\text{new}}$, facilitating sampling.

**Theorem 1** (Takimoto-Warmuth [35]). *Given a DAG $\mathcal{G} = (V, E)$ with designated source node $s \in V$ and sink node $t \in V$, assume $\mathcal{N}$ is the number of paths in $\mathcal{G}$ from $s$ to $t$, $L^*$ is the total loss of best path, and $B$ is an upper bound on the loss of any path in each trial. Then with proper tuning of the learning rate $\eta$ over the $T$ trials, EH guarantees:*

$$\mathbb{E}[L_{EH}] - L^* \leq B \sqrt{2\,T \log \mathcal{N}} + B \log \mathcal{N}.$$

**Component Hedge on Paths** Koolen, Warmuth and Kivinen [25] applied CH to the path problem. The edges are the components of the paths. A path is encoded as a bit vector $\boldsymbol{\pi}$ of $|E|$ components where the 1-bits are the edges in the path. The convex hull of all paths is called the *unit-flow polytope*. CH maintains a mixture vector in this polytope. The constraints of the polytope enforce an outflow of 1 from the source node $s$, and flow conservation at every other node but the sink node $t$. In each trial, the weight of each edge $w_e$ is updated multiplicatively by the factor $\exp(-\eta \ell_e)$. Then the weight vector is projected back to the unit-flow polytope via a relative entropy projection. This projection is achieved by iteratively projecting onto the flow constraint of a particular vertex and then repeatedly cycling through the vertices [8]. Finally, to sample with the same expectation as the mixture vector in the polytope, this vector is decomposed into paths using a greedy approach which removes one path at a time and zeros out at least one edge in the remaining mixture vector in each iteration.

**Theorem 2** (Koolen-Warmuth-Kivinen [25]). *Given a DAG $\mathcal{G} = (V, E)$ with designated source node $s \in V$ and sink node $t \in V$, let $D$ be a length bound of the paths in $\mathcal{G}$ from $s$ to $t$ against which the CH algorithm is compared. Also denote the total loss of the best path of length at most $D$ by $L^*$. Then with proper tuning of the learning rate $\eta$ over the $T$ trials, CH guarantees:*

$$\mathbb{E}[L_{CH}] - L^* \leq D \sqrt{4\,T \log |V|} + 2\,D \log |V|.$$

Much of this paper is concerned with generalizing the tools sketched in this section from paths to $k$-mulitpaths, from the unit-flow polytope to the $k$-flow polytope and developing a generalized version of weight pushing for $k$-multipaths.

## 3 Learning k-Multipaths

As we shall see, $k$-multipaths will be subgraphs of $k$-*DAGs* built from $k$-*multiedges*. Examples of all the definitions are given in Figure 1 for the case $k = 2$.

**Definition 1** ($k$-DAG). *A DAG $\mathcal{G} = (V, E)$ is called $k$-DAG if it has following properties:*

  (i) *There exists one designated "source" node $s \in V$ with no incoming edges.*

 (ii) *There exists a set of "sink" nodes $\mathcal{T} \subset V$ which is the set of nodes with no outgoing edges.*

(iii) *For all non-sink vertices $v$, the set of edges leaving $v$ is partitioned into disjoint sets of size $k$ which are called $k$-multiedges.*

*We denote the set of multiedges "leaving" vertex $v$ as $M_v$ and all multiedges of the DAG as $M$.*

Each $k$-multipath can be generated by starting with a single multiedge at the source and choosing inflow many (i.e. number of incoming edges many) successor multiedges at the internal nodes (until we reach the sink nodes in $\mathcal{T}$). An example of a 2-multipath is given in Figure 1. Recall that paths were described as bit vectors $\boldsymbol{\pi}$ of size $|E|$ where the 1-bits were the edges in the path. In $k$-multipaths each edge bit $\pi_e$ becomes a non-negative count.

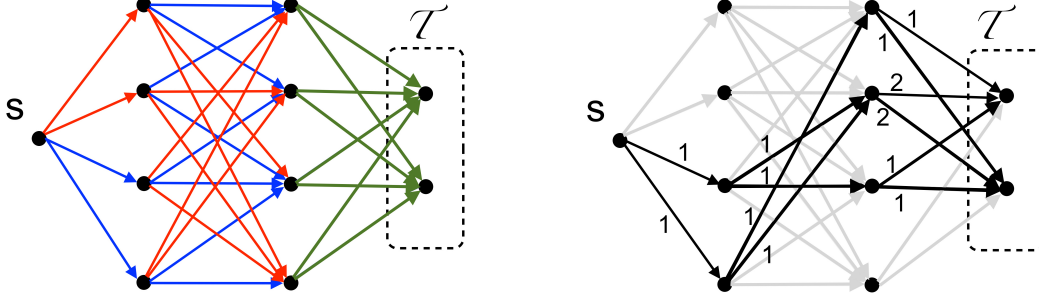

Figure 1: On the left we give an example of a 2-DAG. The source $s$ and the nodes in the first layer each have two 2-multiedges depicted in red and blue. The nodes in the next layer each have one 2-multiedge depicted in green. An example of 2-multipath in the 2-DAG is given on the right. The 2-multipath is represented as an $|E|$-dimensional count vector $\boldsymbol{\pi}$. The grayed edges are the edges with count $\pi_e = 0$. All non-zero counts $\pi_e$ are shown next to their associated edges $e$. Note that for nodes in the middle layers, the outflow is always 2 times the inflow.

**Definition 2** ($k$-multipath). *Given a $k$-DAG $\mathcal{G} = (V, E)$, let $\boldsymbol{\pi} \in \mathbb{N}^{|E|}$ in which $\pi_e$ is associated with $e \in E$. Define the inflow $\pi_{in}(v) := \sum_{(u,v) \in E} \pi_{(u,v)}$ and the outflow $\pi_{out}(v) := \sum_{(v,u) \in E} \pi_{(v,u)}$. We call $\boldsymbol{\pi}$ a $k$-multipath if it has the below properties:*

  (i) *The outflow $\pi_{out}(s)$ of the source $s$ is $k$.*
  (ii) *For any two edges $e, e'$ in a multiedge $m$ of $\mathcal{G}$, $\pi_e = \pi_{e'}$. (When clear from the context, we denote this common value as $\pi_m$.)*
  (iii) *For each vertex $v \in V - \mathcal{T} - \{s\}$, the outflow is $k$ times the inflow, i.e. $\pi_{out}(v) = k \times \pi_{in}(v)$.*

$k$**-Multipath Learning Problem**   We define the problem of *online learning of $k$-multipaths* on a given $k$-DAG as follows. In each trial, the algorithm randomly predicts with a $k$-multipath $\boldsymbol{\pi}$. Then for each edge $e \in E$, the adversary reveals a loss $\ell_e \in [0, 1]$ incurred during that trial. The linear loss of the algorithm during this trial is given by $\boldsymbol{\pi} \cdot \boldsymbol{\ell}$. Observe that the online shortest path problem is a special case when $k = |\mathcal{T}| = 1$. In the remainder of this section, we generalize the algorithms in Section 2 to the online learning problem of $k$-multipaths.

### 3.1   Expanded Hedge on k-Multipaths

We implement EH efficiently for learning $k$-multipath by considering each $k$-multipath as an expert. Recall that each $k$-multipath can be generated by starting with a single multiedge at the source and choosing inflow many successor multiedges at the internal nodes. Multipaths are composed of multiedges as components and *with each multiedge $m \in M$, we associate a weight $w_m$*. We maintain a distribution $W$ over multipaths defined in terms of the weights $\boldsymbol{w} \in \mathbb{R}_{\geq 0}^{|M|}$ on the multiedges. The distribution $W$ will have the following canonical properties:

**Definition 3** (EH distribution properties).

  1. *The weights are in product form, i.e. $W(\boldsymbol{\pi}) = \prod_{m \in M} (w_m)^{\pi_m}$. Recall that $\pi_m$ is the common value in $\boldsymbol{\pi}$ among edges in $m$.*

  2. *The weights are locally normalized, i.e. $\sum_{m \in M_v} w_m = 1$ for all $v \in V - \mathcal{T}$.*

  3. *The total path weight is one, i.e. $\sum_{\boldsymbol{\pi}} W(\boldsymbol{\pi}) = 1$.*

Using these properties, sampling a $k$-multipath from $W$ can be easily done as follows. We start with sampling a single $k$-multiedge at the source and continue sampling inflow many successor multiedges at the internal nodes until the $k$-multipath reaches the sink nodes in $\mathcal{T}$. Observe that $\pi_m$ indicates the number of times the $k$-multiedge $m$ is sampled through this process. EH updates the weights of the

multipaths as follows:

$$W^{\text{new}}(\boldsymbol{\pi}) = \frac{1}{Z} W(\boldsymbol{\pi}) \, \exp(-\eta \, \boldsymbol{\pi} \cdot \boldsymbol{\ell})$$

$$= \frac{1}{Z} \left( \prod_{m \in M} (w_m)^{\pi_m} \right) \exp\left[ -\eta \sum_{m \in M} \pi_m \left( \sum_{e \in m} \ell_e \right) \right]$$

$$= \frac{1}{Z} \prod_{m \in M} \Big( \underbrace{w_m \, \exp\Big[ -\eta \sum_{e \in m} \ell_e \Big]}_{:= \widehat{w}_m} \Big)^{\pi_m}.$$

Thus the weights $w_m$ of each $k$-multiedge $m \in M$ are updated multiplicatively to $\widehat{w}_m$ by multiplying the $w_m$ with the exponentiated loss factors $\exp\left[-\eta \sum_{e \in m} \ell_e\right]$ and then renormalizing with $Z$. Note that $\sum_{e \in m} \ell_e$ is the loss of multiedge $m$.

**Generalized Weight Pushing**   We generalize the *weight pushing algorithm* [31] to $k$-multipaths to reestablish the three canonical properties of Definition 3.   The new weights $W^{\text{new}}(\boldsymbol{\pi}) = \frac{1}{Z} \prod_{m \in M} (\widehat{w}_m)^{\pi_m}$ sum to 1 (i.e.  Property (iii) holds) since $Z$ normalizes the weights.   Our goal is to find new multiedge weights $w_m^{\text{new}}$ so that the other two properties hold as well, i.e. $W^{\text{new}}(\boldsymbol{\pi}) = \prod_{m \in M} (w_m^{\text{new}})^{\pi_m}$ and $\sum_{m \in M_v} w_m^{\text{new}} = 1$ for all nonsinks $v$. For this purpose, we introduce a normalization $Z_v$ for each vertex $v$. Note that $Z_s = Z$ where $s$ is the source node. Now the *generalized weight pushing* finds new weights $w_m^{\text{new}}$ for the multiedges to be used in the next trial:

1. For sinks $v \in \mathcal{T}$, $Z_v := 1$.

2. Recursing backwards in the DAG, let $Z_v := \sum_{m \in M_v} \widehat{w}_m \prod_{u:(v,u) \in m} Z_u$ for all non-sinks $v$.

3. For each multiedge $m$ from $v$ to $u_1, \ldots, u_k$, $w_m^{\text{new}} := \widehat{w}_m \left( \prod_{i=1}^{k} Z_{u_i} \right) / Z_v$.

Appendix B proves the correctness and time complexity of this generalized weight pushing algorithm.

**Regret Bound**   In order to apply the regret bound of EH [16], we have to initialize the distribution $W$ on $k$-multipaths to the uniform distribution. This is achieved by setting all $w_m$ to 1 followed by an application of generalized weight pushing. Note that Theorem 1 is a special case of the below theorem for $k = 1$.

**Theorem 3.** *Given a $k$-DAG $\mathcal{G}$ with designated source node $s$ and sink nodes $\mathcal{T}$, assume $\mathcal{N}$ is the number of $k$-multipaths in $\mathcal{G}$ from $s$ to $\mathcal{T}$, $L^*$ is the total loss of best $k$-multipath, and $B$ is an upper bound on the loss of any $k$-multipath in each trial. Then with proper tuning of the learning rate $\eta$ over the $T$ trials, EH guarantees:*

$$\mathbb{E}[L_{EH}] - L^* \leq B \sqrt{2 \, T \, \log \mathcal{N}} + B \log \mathcal{N}.$$

### 3.2  Component Hedge on k-Multipaths

We implement the CH efficiently for learning of $k$-multipath. Here the $k$-multipaths are the objects which are represented as $|E|$-dimensional[1] count vectors $\boldsymbol{\pi}$ (Definition 2). The algorithm maintains an $|E|$-dimensional mixture vector $\boldsymbol{w}$ in the convex hull of count vectors. This hull is the following polytope over weight vectors obtained by relaxing the integer constraints on the count vectors:

**Definition 4** ($k$-flow polytope)**.** *Given a $k$-DAG $\mathcal{G} = (V, E)$, let $\boldsymbol{w} \in \mathbb{R}_{\geq 0}^{|E|}$ in which $w_e$ is associated with $e \in E$. Define the inflow $w_{in}(v) := \sum_{(u,v) \in E} w_{(u,v)}$ and the outflow $w_{out}(v) := \sum_{(v,u) \in E} w_{(v,u)}$. $\boldsymbol{w}$ belongs to the $k$-flow polytope of $\mathcal{G}$ if it has the below properties:*

(i) *The outflow $w_{out}(s)$ of the source $s$ is $k$.*
(ii) *For any two edges $e, e'$ in a multiedge $m$ of $\mathcal{G}$, $w_e = w_{e'}$.*
(iii) *For each vertex $v \in V - \mathcal{T} - \{s\}$, the outflow is $k$ times the inflow, i.e. $w_{out}(v) = k \times w_{in}(v)$.*

In each trial, the weight of each edge $w_e$ is updated multiplicatively to $\widehat{w}_e = w_e \exp(-\eta \ell_e)$ and then the weight vector $\widehat{\boldsymbol{w}}$ is projected back to the $k$-flow polytope via a relative entropy projection:

$$\boldsymbol{w}^{\text{new}} := \underset{\boldsymbol{w} \in k\text{-flow polytope}}{\arg\min} \Delta(\boldsymbol{w}||\widehat{\boldsymbol{w}}), \quad \text{where} \quad \Delta(\boldsymbol{a}||\boldsymbol{b}) = \sum_i a_i \log \frac{a_i}{b_i} + b_i - a_i.$$

This projection is achieved by repeatedly cycling over the vertices and enforcing the local flow constraints at the current vertex. Based on the properties of the $k$-flow polytope in Definition 4, the corresponding projection steps can be rewritten as follows:

(i) Normalize the $w_{\text{out}}(s)$ to $k$.

(ii) Given a multiedge $m$, set the $k$ weights in $m$ to their geometric average.

(iii) Given a vertex $v \in V - \mathcal{T} - \{s\}$, scale the adjacent edges of $v$ s.t.

$$w_{\text{out}}(v) := \sqrt[k+1]{k \, (w_{\text{out}}(v))^k \, w_{\text{in}}(v)} \quad \text{and} \quad w_{\text{in}}(v) := \frac{1}{k} \sqrt[k+1]{k \, (w_{\text{out}}(v))^k \, w_{\text{in}}(v)}.$$

See Appendix C for details.

**Decomposition** The flow polytope has exponentially many objects as its corners. We now rewrite any vector $\boldsymbol{w}$ in the polytope as a mixture of $|M|$ objects. CH then predicts with a random object drawn from this sparse mixture. The mixture vector is decomposed by greedily removing a multipath from the current weight vector as follows: Ignore all edges with zero weights. Pick a multiedge at $s$ and iteratively inflow many multiedges at the internal nodes until you reach the sink nodes. Now subtract that constructed multipath from the mixture vector $\boldsymbol{w}$ scaled by its minimum edge weight. This zeros out at least $k$ edges and maintain the flow constraints at the internal nodes.

**Regret Bound** The regret bound for CH depends on a good choice of the initial weight vector $\boldsymbol{w}^{\text{init}}$ in the $k$-flow polytope. We use an initialization technique recently introduced in [32]. Instead of explicitly selecting $\boldsymbol{w}^{\text{init}}$ in the $k$-flow polytope, the initial weight is obtained by projecting a point $\widehat{\boldsymbol{w}}^{\text{init}}$ outside of the polytope to the inside. This yields the following regret bounds (Appendix D):

**Theorem 4.** *Given a $k$-DAG $\mathcal{G} = (V, E)$, let $D$ be the upper bound for the 1-norm of the $k$-multipaths in $\mathcal{G}$. Also denote the total loss of the best $k$-multipath by $L^*$. Then with proper tuning of the learning rate $\eta$ over the $T$ trials, CH guarantees:*

$$\mathbb{E}[L_{CH}] - L^* \leq D \sqrt{2 \, T \, (2 \log |V| + \log D)} + 2 \, D \, \log |V| + D \log D.$$

*Moreover, when the $k$-multipaths are bit vectors, then:*

$$\mathbb{E}[L_{CH}] - L^* \leq D \sqrt{4 \, T \, \log |V|} + 2 \, D \, \log |V|.$$

Notice that by setting $|\mathcal{T}| = k = 1$, the algorithm for path learning in [25] is recovered. Also observe that Theorem 2 is a corollary of Theorem 4 since every path is represented as a bit vector.

## 4 Online Dynamic Programming with Multipaths

We consider the problem of repeatedly solving a variant of the same dynamic programming problem in successive trials. We will use our definition of $k$-DAGs to describe a certain type of dynamic programming problem. The vertex set $V$ is a set of subproblems to be solved. The source node $s \in V$ is the final subproblem. The sink nodes $\mathcal{T} \subset V$ are the base subproblems. An edge from a node $v$ to another node $v'$ means that subproblem $v$ may recurse on $v'$. We assume a non-base subproblem $v$ always breaks into exactly $k$ smaller subproblems. A step of the dynamic programming recursion is thus represented by a $k$-multiedge. We assume the sets of $k$ subproblems between possible recursive calls at a node are disjoint. This corresponds to the fact that the choice of multiedges at a node partitions the edge set leaving that node.

There is a loss associated with any sink node in $\mathcal{T}$. Also with the recursions at the internal node $v$ a local loss will be added to the loss of the subproblems that depends on $v$ and the chosen $k$-multiedge

leaving $v$. Recall that $M_v$ is the set of multiedges leaving $v$. We can handle the following type of "min-sum" recurrences:

$$\text{OPT}(v) = \begin{cases} L_{\mathcal{T}}(v) & v \in \mathcal{T} \\ \min_{m \in M_v} \left[ \sum_{u:(v,u) \in m} \text{OPT}(u) + L_M(m) \right] & v \in V - \mathcal{T}. \end{cases}$$

The problem of repeatedly solving such a dynamic programming problem over trials now becomes the problem of online learning of $k$-multipaths in this $k$-DAG. Note that due to the correctness of the dynamic programming, every possible solution to the dynamic programming can be encoded as a $k$-multipath in the $k$-DAG and vice versa.

The loss of a given multipath is the sum of $L_M(m)$ over all multiedges $m$ in the multipath plus the sum of $L_{\mathcal{T}}(v)$ for all sink nodes $v$ at the bottom of the multipath. To capture the same loss, we can alternatively define losses over the edges of the $k$-DAG. Concretely, for each edge $(v, u)$ in a given multiedge $m$ define $\ell_{(v,u)} := \frac{1}{k} L_M(m) + \mathbb{1}_{\{u \in \mathcal{T}\}} L_{\mathcal{T}}(u)$ where $\mathbb{1}_{\{\cdot\}}$ is the indicator function.

In summary we are addressing the above min-sum type dynamic programming problem specified by a $k$-DAG and local losses where for the sake of simplicity we made two assumptions: each non-base subproblem breaks into exactly $k$ smaller subproblems and the choice of $k$ subproblems at a node are disjoint. We briefly discuss in the conclusion section how to generalize our methods to arbitrary min-sum dynamic programming problems, where the sets of subproblems can overlap and may have different sizes.

## 4.1 The Example of Learning Binary Search Trees

Recall again the online version of optimal binary search tree (BST) problem [10]: We are given a set of $n$ distinct keys $K_1 < K_2 < \ldots < K_n$ and $n+1$ gaps or "dummy keys" $D_0, \ldots, D_n$ indicating *search failures* such that for all $i \in \{1..n\}$, $D_{i-1} < K_i < D_i$. In each trial, the algorithm predicts with a BST. Then the adversary reveals a frequency vector $\boldsymbol{\ell} = (\boldsymbol{p}, \boldsymbol{q})$ with $\boldsymbol{p} \in [0,1]^n$, $\boldsymbol{q} \in [0,1]^{n+1}$ and $\sum_{i=1}^n p_i + \sum_{j=0}^n q_j = 1$. For each $i, j$, the frequencies $p_i$ and $q_j$ are the *search probabilities* for $K_i$ and $D_j$, respectively. The loss is defined as the average search cost in the predicted BST which is the average depth[2] of all the nodes in the BST:

$$\text{loss} = \sum_{i=1}^n \text{depth}(K_i) \cdot p_i + \sum_{j=0}^n \text{depth}(D_j) \cdot q_j.$$

**Convex Hull of BSTs**   Implementing CH requires a representation where not only the BST polytope has a polynomial number of facets, but also the loss must be linear over the components. Since the average search cost is linear in the $\text{depth}(K_i)$ and $\text{depth}(D_j)$ variables, it would be natural to choose these $2n+1$ variables as the components for representing a BST. Unfortunately the convex hull of all BSTs when represented this way is not known to be a polytope with a polynomial number of facets. There is an alternate characterization of the convex hull of BSTs with $n$ internal nodes called the *associahedron* [29]. This polytope has polynomial in $n$ many facets but the average search cost is not linear in the $n$ components associated with this polytope[3].

**The Dynamic Programming Representation**   The optimal BST problem can be solved via dynamic programming [10]. Each subproblem is denoted by a pair $(i, j)$, for $1 \le i \le n+1$ and $i - 1 \le j \le n$, indicating the optimal BST problem with the keys $K_i, \ldots, K_j$ and dummy keys $D_{i-1}, \ldots, D_j$. The base subproblems are $(i, i-1)$, for $1 \le i \le n+1$ and the final subproblem is $(1, n)$. The BST dynamic programming problem uses the following recurrence:

$$\text{OPT}(i,j) = \begin{cases} q_{i-1} & j = i-1 \\ \min_{i \le r \le j} \{ \text{OPT}(i, r-1) + \text{OPT}(r+1, j) + \sum_{k=i}^j p_k + \sum_{k=i-1}^j q_k \} & i \le j. \end{cases}$$

This recurrence always recurses on 2 subproblems. Therefore we have $k = 2$ and the associated 2-DAG has the subproblems/vertices $V = \{(i,j) | 1 \le i \le n+1, i - 1 \le j \le n\}$, source $s = (1, n)$

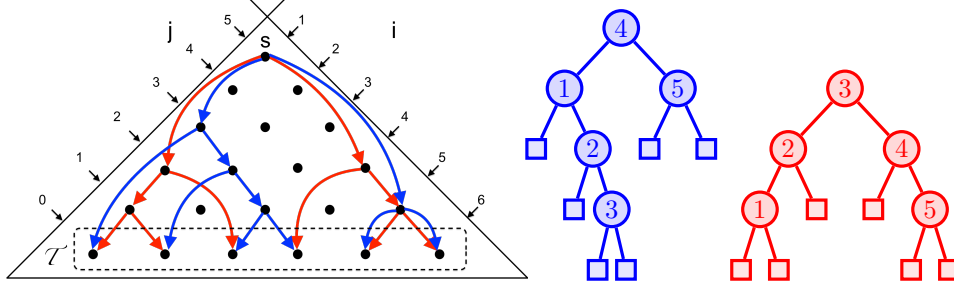

Figure 2: (left) Two different 2-multipaths in the DAG, in red and blue, and (right) their associated BSTs of $n = 5$ keys and 6 "dummy" keys. Note that each node, and consequently edge, is visited at most once in these 2-multipaths.

| Problem | FPL | EH | CH |
|---|---|---|---|
| Optimal Binary Search Trees | $\mathcal{O}(n^{\frac{3}{2}}\sqrt{T})$ | $\mathcal{O}(n^{\frac{3}{2}}\sqrt{T})$ | $\mathcal{O}(n\,(\log n)^{\frac{1}{2}}\sqrt{T})$ |
| Matrix-Chain Multiplications[4] | — | $\mathcal{O}(n^{\frac{3}{2}}\,(d_{\max})^3\,\sqrt{T})$ | $\mathcal{O}(n\,(\log n)^{\frac{1}{2}}\,(d_{\max})^3\,\sqrt{T})$ |
| Knapsack | $\mathcal{O}(n^{\frac{3}{2}}\sqrt{T})$ | $\mathcal{O}(n^{\frac{3}{2}}\sqrt{T})$ | $\mathcal{O}(n\,(\log nC)^{\frac{1}{2}}\sqrt{T})$ |
| Rod Cutting | $\mathcal{O}(n^{\frac{3}{2}}\sqrt{T})$ | $\mathcal{O}(n^{\frac{3}{2}}\sqrt{T})$ | $\mathcal{O}(n\,(\log n)^{\frac{1}{2}}\sqrt{T})$ |
| Weighted Interval Scheduling | $\mathcal{O}(n^{\frac{3}{2}}\sqrt{T})$ | $\mathcal{O}(n^{\frac{3}{2}}\sqrt{T})$ | $\mathcal{O}(n\,(\log n)^{\frac{1}{2}}\sqrt{T})$ |

Table 1: Performance of various algorithms over different problems. $C$ is the capacity in the Knapsack problem, and $d_{\max}$ is the upper-bound on the dimension in matrix-chain multiplication problem.

and sinks $\mathcal{T} = \{(i, i-1)|1 \le i \le n+1\}$. Also at node $(i, j)$, the set $M_{(i,j)}$ consists of $(j - i + 1)$ many 2-multiedges. The $r$th 2-multiedge leaving $(i, j)$ comprised of 2 edges going from the node $(i, j)$ to the nodes $(i, r-1)$ and $(r+1, j)$. Figure 2 illustrates the 2-DAG and 2-multipaths associated with BSTs.

Since the above recurrence relation correctly solves the offline optimization problem, every 2-multipath in the DAG represents a BST, and every possible BST can be represented by a 2-multipath of the 2-DAG. We have $O(n^3)$ edges and multiedges which are the components of our new representation. The loss of each 2-multiedge leaving $(i, j)$ is $\sum_{k=i}^{j} p_k + \sum_{k=i-1}^{j} q_k$ and is upper bounded by 1. Most crucially, the original average search cost is linear in the losses of the multiedges and the 2-flow polytope has $O(n^3)$ facets.

**Regret Bound**  As mentioned earlier, the number of binary trees with $n$ nodes is the $n$th Catalan number. Therefore $\mathcal{N} = \frac{(2n)!}{n!(n+1)!} \in (2^n, 4^n)$. Also note that the expected search cost is bounded by $B = n$ in each trial. Thus using Theorem 3, EH achieves a regret bound of $\mathcal{O}(n^{\frac{3}{2}}\sqrt{T})$.

Additionally, notice that the number of subproblems in the dynamic programming problem for BSTs is $\frac{(n+1)(n+2)}{2}$. This is also the number of vertices in the associated 2-DAG and each 2-multipath representing a BST consists of exactly $D = 2n$ edges. Therefore using Theorem 4, CH achieves a regret bound of $\mathcal{O}(n\,(\log n)^{\frac{1}{2}}\sqrt{T})$.

## 5    Conclusions and Future Work

We developed a general framework for online learning of combinatorial objects whose offline optimization problems can be efficiently solved via an algorithm belonging to a large class of dynamic programming algorithms. In addition to BSTs, several example problems are discussed in Appendix A. Table 1 gives the performance of EH and CH in our dynamic programming framework

and compares it with the Follow the Perturbed Leader (FPL) algorithm. FPL additively perturbs the losses and then uses dynamic programming to find the solution of minimum loss. FPL essentially always matches EH, and CH is better than both in all cases.

We conclude with a few remarks:

- For EH, projections are simply a renormalization of the weight vector. In contrast, iterative Bregman projections are often needed for projecting back into the polytope used by CH [25, 19]. These methods are known to converge to the exact projection [8, 6] and are reported to be very efficient empirically [25]. For the special cases of Euclidean projections [13] and Sinkhorn Balancing [24], linear convergence has been proven. However we are unaware of a linear convergence proof for general Bregman divergences. Regardless of the convergence rate, the remaining gaps to the exact projections have to be accounted for as additional loss in the regret bounds. We do this in Appendix E for CH.

- For the sake of concreteness, we focused in this paper on dynamic programming problems with "min-sum" recurrence relations, a fixed branching factor $k$ and mutually exclusive sets of choices at a given subproblem. However, our results can be generalized to arbitrary "min-sum" dynamic programming problems with the methods introduced in [30]: We let the multiedges in $\mathcal{G}$ form *hyperarcs*, each of which is associated with a loss. Furthermore, each combinatorial object is encoded as a *hyperpath*, which is a sequence of hyperarcs from the source to the sinks. The polytope associated with such a dynamic programming problem is defined by flow-type constraints over the underlying *hypergraph* $\mathcal{G}$ of subproblems. Thus online learning a dynamic programming solution becomes a problem of learning hyperpaths in a hypergraph, and the techniques introduced in this paper let us implement EH and CH for this more general class of dynamic programming problems.

- In this work we use dynamic programming algorithms for building polytopes for combinatorial objects that have a polynomial number of facets. The technique of going from the original polytope to a higher dimensional polytope in order to reduce the number of facets is known as *extended formulation* (see e.g. [21]). In the learning application we also need the additional requirement that the loss is linear in the components of the objects. A general framework of using extended formulations to develop learning algorithms has recently been explored in [32].

- We hope that many of the techniques from the expert setting literature can be adapted to learning combinatorial objects that are composed of components. This includes lower bounding weights for shifting comparators [20] and sleeping experts [7, 1]. Also in this paper, we focus on *full information* setting where the adversary reveals the entire loss vector in each trial. In contrast in *full-* and *semi-bandit* settings, the adversary only reveals partial information about the loss. Significant work has already been done in learning combinatorial objects in full- and semi-bandit settings [3, 18, 4, 27, 9]. It seems that the techniques introduced in the paper will also carry over.

- *Online Markov Decision Processes (MDPs)* [15, 14] is an online learning model that focuses on the sequential revelation of an object using a sequential state based model. This is very much related to learning paths and the sequential decisions made in our dynamic programming framework. Connecting our work with the large body of research on MDPs is a promising direction of future research.

- There are several important dynamic programming instances that are not included in the class considered in this paper: The Viterbi algorithm for finding the most probable path in a graph, and variants of Cocke-Younger-Kasami (CYK) algorithm for parsing probabilistic context-free grammars. The solutions for these problems are min-sum type optimization problem after taking a log of the probabilities. However taking logs creates unbounded losses. Extending our methods to these dynamic programming problems would be very worthwhile.

**Acknowledgments** We thank S.V.N. Vishwanathan for initiating and guiding much of this research. We also thank Michael Collins for helpful discussions and pointers to the literature on hypergraphs and PCFGs. This research was supported by the National Science Foundation (NSF grant IIS-1619271).

## Footnotes

[1]For convenience we use the edges as components for CH instead of the multiedges as for EH.

[2]Here the root starts at depth 1.

[3]Concretely, the $i$th component is $a_i b_i$ where $a_i$ and $b_i$ are the number of nodes in the left and right subtrees of the $i$th internal node $K_i$, respectively.

[4]The loss of a fully parenthesized matrix-chain multiplication is the number of scalar multiplications in the execution of all matrix products. This number cannot be expressed as a linear loss over the dimensions of the matrices. We are thus unaware of a way to apply FPL to this problem using the dimensions of the matrices as the components. See Appendix A.1 for more details.

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
