[Supplementary Material]

# A More Instantiations

## A.1 Matrix Chain Multiplication

Given a sequence $A_1, A_2, \ldots, A_n$ of $n$ matrices, our goal is to compute the product $A_1 \times A_2 \times \ldots \times A_n$ in the most efficient way. Using the standard algorithm for multiplying pairs of matrices as a subroutine, this product can be found by a specifying the order which the matrices are multiplied together. This order is determined by a *full parenthesization*: A product of matrices is fully parenthesized if it is either a single matrix or the multiplication of two fully parenthesized matrix products surrounded by parentheses. For instance, there are five full parenthesizations of the product $A_1 A_2 A_3 A_4$:

$$(A_1(A_2(A_3 A_4)))$$
$$(A_1((A_2 A_3)A_4))$$
$$((A_1 A_2)(A_3 A_4))$$
$$(((A_1 A_2)A_3)A_4)$$
$$((A_1(A_2 A_3))A_4).$$

We consider the online version of *matrix-chain multiplication* problem [10]. In each trial, the algorithm predicts with a full parenthesization of the product $A_1 \times A_2 \times \ldots \times A_n$ without knowing the dimensions of these matrices. Then the adversary reveals the dimensions of each $A_i$ at the end of the trial denoted by $d_{i-1} \times d_i$ for all $i \in \{1..n\}$. The loss of the algorithm is defined as the number of scalar multiplications in the matrix-chain product in that trial. The goal is to predict with a sequence of full parenthesizations minimizing regret which is the difference between the total loss of the algorithm and the total loss of the single best full parenthesization chosen in hindsight.

The number of scalar multiplications in the matrix-chain product cannot be expressed as a linear loss over the dimensions of the matrices $d_i$'s. Thus we are unaware of a way to apply FPL to this problem using the $d_i$'s as components in the loss vector revealed by the adversary.

**The Dynamic Programming Representation** Finding the best full parenthesization can be solved via dynamic programming [10]. Each subproblem is denoted by a pair $(i, j)$ for $1 \le i \le j \le n$, indicating the problem of finding a full parenthesization of the partial matrix product $A_i \ldots A_j$. The base subproblems are $(i, i)$ for $1 \le i \le n$ and the final subproblem is $(1, n)$. The dynamic programming for matrix chain multiplication uses the following recurrence:

$$\text{OPT}(i, j) = \begin{cases} 0 & i = j \\ \min_{i \le k < j}\{\text{OPT}(i, k) + \text{OPT}(k+1, j) + d_{i-1}\, d_k\, d_j\} & i < j. \end{cases}$$

This recurrence always recurses on 2 subproblems. Therefore we have $k = 2$ and the associated 2-DAG has the subproblems/vertices $V = \{(i, j) \mid 1 \le i \le j \le n\}$, source $s = (1, n)$ and sinks $\mathcal{T} = \{(i, i) \mid 1 \le i \le n\}$. Also at node $(i, j)$, the set $M_{(i,j)}$ consists of $(j - i)$ many 2-multiedges. The $k$th 2-multiedge leaving $(i, j)$ is comprised of 2 edges going from the node $(i, j)$ to the nodes $(i, k)$ and $(k + 1, j)$. The loss of the $k$th 2-multiedge is $d_{i-1}\, d_k\, d_j$. Figure 3 illustrates the 2-DAG and 2-multipaths associated with matrix chain multiplications.

Since the above recurrence relation correctly solves the offline optimization problem, every 2-multipath in the DAG represents a full parenthesization, and every possible full parenthesization can be represented by a 2-multipath of the 2-DAG.

We have $O(n^3)$ edges and multiedges which are the components of our new representation.

Assuming that all dimensions $d_i$ are bounded as $d_i < d_{\max}$ for some $d_{\max}$, the loss associated with each 2-multiedge is upper-bounded by $(d_{\max})^3$. Most crucially, the original number of scalar multiplications in the matrix-chain product is linear in the losses of the multiedges and the 2-flow polytope has $O(n^3)$ facets.

**Regret Bounds** It is well-known that the number of full parenthesizations of a sequence of $n$ matrices is the $n$th Catalan number [10]. Therefore $\mathcal{N} = \frac{(2n)!}{n!(n+1)!} \in (2^n, 4^n)$. Also note that the number of scalar multiplications in each full parenthesization is bounded by $B = (n-1)(d_{\max})^3$ in each trial. Thus using Theorem 3, EH achieves a regret bound of $\mathcal{O}(n^{\frac{3}{2}}\,(d_{\max})^3\,\sqrt{T})$.

Figure 3: Given a chain of $n = 4$ matrices, the 2-multipaths associated with the full parenthesizations $((A_1 A_2)(A_3 A_4))$ and $(A_1((A_2 A_3)A_4))$ are depicted in red and blue, respectively.

Additionally, notice that each 2-multipath associated with a full parenthesization consists of exactly $D = 2(n-1)$ edges. Also we have $|V| = \frac{n(n+1)}{2}$. Therefore, incorporating $(d_{\max})^3$ as the loss range for each component and using Theorem 4 , CH achieves a regret bound of $\mathcal{O}(n (\log n)^{\frac{1}{2}} (d_{\max})^3 \sqrt{T})$.

## A.2 Knapsack

Consider the online version of the *knapsack* problem [23]: We are given a set of $n$ items along with the *capacity* of the knapsack $C \in \mathbb{N}$. For each item $i \in \{1..n\}$, a *heaviness* $h_i \in \mathbb{N}$ is associated. In each trial, the algorithm predicts with a *packing* which is a subset of items whose total heaviness is at most the capacity of the knapsack. After the prediction of the algorithm, the adversary reveals the profit of each item $p_i \in [0, 1]$. The gain is defined as the sum of the profits of the items picked in the packing predicted by the algorithm in that trial. The goal is to predict with a sequence of packings minimizing regret which is the difference between the total gain of the algorithm and the total gain of the single best packing chosen in hindsight.

Note that this online learning problem only deals with exponentially many objects when there are exponentially many feasible packings. If the number of packings is polynomial, then it is practical to simply run the Hedge algorithm with one weight per packing. Here we consider a setting of the problem where maintaining one weight per packing is impractical. We assume $C$ and $h_i$'s are in such way that the number of feasible packings is exponential in $n$.

**The Dynamic Programming Representation**   Finding the optimal packing can be solved via dynamic programming [23]. Each subproblem is denoted by a pair $(i, c)$ for $0 \leq i \leq n$ and $0 \leq c \leq C$, indicating the knapsack problem given items $1, \ldots, i$ and capacity $c$. The base subproblems are $(0, c)$ for $0 \leq c \leq C$ and the final subproblem is $(n, C)$. The dynamic programming for the knapsack problem uses the following recurrence:

$$\text{OPT}(i, c) = \begin{cases} 0 & i = 0 \\ \text{OPT}(i-1, c) & c < h_i \\ \max\{\text{OPT}(i-1, c), \; p_i + \text{OPT}(i-1, c - h_i)\} & \text{else.} \end{cases}$$

This recurrence always recurses on 1 subproblem. Therefore we have $k = 1$ and the problem is essentially the online longest-path problem with several sink nodes. The associated DAG has the subproblems/vertices $V = \{(i, c) \mid 0 \leq i \leq n, \quad 0 \leq c \leq C\}$, source $s = (n, C)$ and sinks $\mathcal{T} = \{(0, c) \mid 0 \leq c \leq C\}$. Also at node $(i, c)$, the set $M_{(i,c)}$ consists of two edges going from the node $(i, c)$ to the nodes $(i-1, c)$ and $(i-1, c - h_i)$. Figure 4 illustrates the DAG and paths associated with packings.

Since the above recurrence relation correctly solves the offline optimization problem, every path in the DAG represents a packing, and every possible packing can be represented by a path of the DAG.

We have $O(nC)$ edges which are the components of our new representation. The gains of the edges going from the node $(i, c)$ to the nodes $(i-1, c)$ and $(i-1, c - h_i)$ are 0 and $p_i$, respectively. Note that the gain associated with each edge is upper-bounded by 1. Most crucially, the sum of the profits

Figure 4: An example with $C = 7$ and $(h_1, h_2, h_3) = (2, 3, 4)$. The packing of picking the first and third item is highlighted.

of the picked items in the packing is linear in the gains of the edges and the unit-flow polytope has $O(n C)$ facets.

**Regret Bounds**    We turn the problem into shortest-path problem by defining a loss for each edge $e \in E$ as $\ell_e = 1 - g_e$ in which $g_e$ is the gain of $e$. Call this new DAG $\bar{\mathcal{G}}$. Let $L_{\bar{\mathcal{G}}}(\pi)$ be the loss of path $\pi$ in $\bar{\mathcal{G}}$ and $G_{\mathcal{G}}(\pi)$ be the gain of path $\pi$ in $\mathcal{G}$. Since all paths contain exactly $n$ edges, the loss and gain are related as follows: $L_{\bar{\mathcal{G}}}(\boldsymbol{\pi}) = n - G_{\mathcal{G}}(\boldsymbol{\pi})$.

According to our initial assumption $\log \mathcal{N} = \mathcal{O}(n)$. Also note that loss of each path in each trial is bounded by $B = n$. Thus using Theorem 3 we obtain:

$$G^* - \mathbb{E}[G_{\text{EH}}] = (nT - L^*) - (nT - \mathbb{E}[L_{\text{EH}}]) = \mathbb{E}[L_{\text{EH}}] - L^*$$
$$= \mathcal{O}(n^{\frac{3}{2}} \sqrt{T}).$$

Notice that the number of vertices is $|V| = nC$ and each path consists of $D = n$ edges. Therefore using Theorem 4 we obtain:

$$G^* - \mathbb{E}[G_{\text{CH}}] = (nT - L^*) - (nT - \mathbb{E}[L_{\text{CH}}]) = \mathbb{E}[L_{\text{CH}}] - L^*$$
$$= \mathcal{O}(n \, (\log nC)^{\frac{1}{2}} \sqrt{T}).$$

## A.3   Rod Cutting

Consider the online version of *rod cutting* problem [10]: A rod of length $n \in \mathbb{N}$ is given. In each trial, the algorithm predicts with a *cutting*, that is, it cuts up the rod into smaller pieces of integer length. Then the adversary reveals a *profit* $p_i \in [0, 1]$ for each piece of length $i \in \{1..n\}$ that can be possibly generated out of a cutting. The gain of the algorithm is defined as the sum of the profits of all the pieces generated by the predicted cutting in that trial. The goal is to predict with a sequence of cuttings minimizing regret which is the difference between the total gain of the algorithm and the total gain of the single best cutting chosen in hindsight. See Figure 5 as an example.

**The Dynamic Programming Representation**    Finding the optimal cutting can be solved via dynamic programming [10]. Each subproblem is simply denoted by $i$ for $0 \leq i \leq n$, indicating the rod cutting problem given a rod of length $i$. The base subproblem is $i = 0$, and the final subproblem is $i = n$. The dynamic programming for the rod cutting problem uses the following recurrence:

$$\text{OPT}(i) = \begin{cases} 0 & i = 0 \\ \max_{0 \leq j \leq i}\{OPT(j) + p_{i-j}\} & i > 0. \end{cases}$$

This recurrence always recurses on 1 subproblem. Therefore we have $k = 1$ and the problem is essentially the online longest-path problem from the source to the sink. The associated DAG has the

Figure 5: All cuttings of a rod of length $n = 4$ and their profits given $(p_1, p_2, p_3, p_4) = (.1, .4, .7, .9)$.

Figure 6: An example of rod cutting problem with $n = 4$. The cutting with two smaller pieces of size 2 is highlighted.

subproblems/vertices $V = \{0, 1, \ldots, n\}$, source $s = n$ and sink $\mathcal{T} = \{0\}$. Also at node $i$, the set $M_i$ consists of $i$ edges going from the node $i$ to the nodes $0, 1, \ldots, i-1$. Figure 6 illustrates the DAG and paths associated with the cuttings.

Since the above recurrence relation correctly solves the offline optimization problem, every path in the DAG represents a cutting, and every possible cutting can be represented by a path of the DAG.

We have $O(n^2)$ edges which are the components of our new representation. The gains of the edges going from the node $i$ to the node $j$ (where $j < i$) is $p_{i-j}$. Note that the gain associated with each edge is upper-bounded by 1. Most crucially, the sum of the profits of all the pieces generated by the cutting is linear in the gains of the edges and the unit-flow polytope has $O(n)$ facets.

**Regret Bounds**  Similar to the knapsack problem, we turn this problem into a shortest-path problem: We first modify the graph so that all paths have equal length of $n$ (which is the length of the longest path) and the gain of each path remains fixed. We apply a method introduced in György et. al. [18], which adds $\mathcal{O}(n^2)$ vertices and edges (with gain zero) to make all paths have the same length. Then we define a loss for each edge $e$ as $\ell_e = 1 - g_e$ in which $g_e$ is the gain of $e$. Call this new DAG $\bar{\mathcal{G}}$. Similar to the knapsack problem, we have $L_{\bar{\mathcal{G}}}(\boldsymbol{\pi}) = n - G_{\mathcal{G}}(\boldsymbol{\pi})$ for all paths $\boldsymbol{\pi}$.

Note that in both $\mathcal{G}$ and $\bar{\mathcal{G}}$, there are $\mathcal{N} = 2^{n-1}$ paths. Also note that loss of each path in each trial is bounded by $B = n$. Thus using Theorem 3 we obtain[5]

$$G^* - \mathbb{E}[G_{\text{EH}}] = (nT - L^*) - (nT - \mathbb{E}[L_{\text{EH}}]) = \mathbb{E}[L_{\text{EH}}] - L^*$$
$$= \mathcal{O}(n^{\frac{3}{2}}\sqrt{T}).$$

Figure 7: An example of weighted interval scheduling with $n = 6$

Notice that the number of vertices in $\bar{\mathcal{G}}$ is $\mathcal{O}(n^2)$ and each path consists of $D = n$ edges. Therefore using Theorem 4 we obtain:

$$G^* - \mathbb{E}[G_{\text{CH}}] = (nT - L^*) - (nT - \mathbb{E}[L_{\text{CH}}]) = \mathbb{E}[L_{\text{CH}}] - L^*$$
$$= \mathcal{O}(n (\log n)^{\frac{1}{2}} \sqrt{T}).$$

### A.4 Weighted Interval Scheduling

Consider the online version of *weighted interval scheduling* problem [23]: We are given a set of $n$ intervals $I_1, \ldots, I_n$ on the real line. In each trial, the algorithm predicts with a *scheduling* which is a subset of non-overlapping intervals. Then, for each interval $I_j$, the adversary reveals $p_j \in [0, 1]$ which is the *profit* of including $I_j$ in the scheduling. The gain of the algorithm is defined as the total profit over chosen intervals in the scheduling in that trial. The goal is to predict with a sequence of schedulings minimizing regret which is the difference between the total gain of the algorithm and the total gain of the single best scheduling chosen in hindsight. See Figure 7 as an example. Note that this problem is only interesting when there are exponential in $n$ many combinatorial objects (schedulings).

**The Dynamic Programming Representation**  Finding the optimal scheduling can be solved via dynamic programming [23]. Each subproblem is simply denoted by $i$ for $0 \le i \le n$, indicating the weighted scheduling problem for the intervals $I_1, \ldots, I_i$. The base subproblem is $i = 0$, and the final subproblem is $i = n$. The dynamic programming for the rod cutting problem uses the following recurrence:

$$\text{OPT}(i) = \begin{cases} 0 & i = 0 \\ \max\{\text{OPT}(i-1), \text{OPT}(\text{pred}(i)) + p_i\} & i > 0. \end{cases}$$

where

$$\text{pred}(i) := \begin{cases} 0 & i = 1 \\ \max_{\{j < i,\, I_i \cap I_j = \emptyset\}} j & i > 1. \end{cases}$$

This recurrence always recurses on 1 subproblem. Therefore we have $k = 1$ and the problem is essentially the online longest-path problem from the source to the sink. The associated DAG has the subproblems/vertices $V = \{0, 1, \ldots, n\}$, source $s = n$ and sink $\mathcal{T} = \{0\}$. Also at node $i$, the set $M_i$ consists of 2 edges going from the node $i$ to the nodes $i - 1$ and $\text{pred}(i)$. Figure 8 illustrates the DAG and paths associated with the scheduling for the example given in Figure 7.

Since the above recurrence relation correctly solves the offline optimization problem, every path in the DAG represents a scheduling, and every possible scheduling can be represented by a path of the DAG.

Figure 8: The underlying DAG associated with the example illustrated in Figure 7. The scheduling with $I_1$, $I_3$, and $I_5$ is highlighted.

We have $O(n)$ edges which are the components of our new representation. The gains of the edges going from the node $i$ to the nodes $i-1$ and $\mathrm{pred}(i)$ are 0 and $p_i$, respectively. Note that the gain associated with each edge is upper-bounded by 1. Most crucially, the total profit over chosen intervals in the scheduling is linear in the gains of the edges and the unit-flow polytope has $O(n)$ facets.

**Regret Bounds**   Similar to rod cutting, this is also the online longest-path problem with one sink node. Like the rod cutting problem, we modify the graph by adding $\mathcal{O}(n^2)$ vertices and edges (with gain zero) to make all paths have the same length and change the gains into losses. Call this new DAG $\bar{\mathcal{G}}$. Again we have $L_{\bar{\mathcal{G}}}(\boldsymbol{\pi}) = n - G_{\mathcal{G}}(\boldsymbol{\pi})$ for all paths $\boldsymbol{\pi}$.

According to our initial assumption $\log \mathcal{N} = \mathcal{O}(n)$. Also note that loss of each path in each trial is bounded by $B = n$. Thus using Theorem 3 we obtain:

$$G^* - \mathbb{E}[G_{\mathrm{EH}}] = (nT - L^*) - (nT - \mathbb{E}[L_{\mathrm{EH}}]) = \mathbb{E}[L_{\mathrm{EH}}] - L^*$$
$$= \mathcal{O}(n^{\frac{3}{2}} \sqrt{T}).$$

Notice that the number of vertices in $\bar{\mathcal{G}}$ is $\mathcal{O}(n^2)$ and each path consists of $D = n$ edges. Therefore using Theorem 4 we obtain:

$$G^* - \mathbb{E}[G_{\mathrm{CH}}] = (nT - L^*) - (nT - \mathbb{E}[L_{\mathrm{CH}}]) = \mathbb{E}[L_{\mathrm{CH}}] - L^*$$
$$= \mathcal{O}(n (\log n)^{\frac{1}{2}} \sqrt{T}).$$

# B   Generalized Weight Pushing

**Lemma 5.** *The weights $w_m^{new}$ generated by the generalized weight pushing satisfies the EH distribution properties in Definition 3 and $W^{new}(\boldsymbol{\pi}) = \frac{1}{Z} \prod_{m \in M} (\widehat{w}_m)^{\pi_m}$. Moreover, the weights $w_m^{new}$ can be computed in $\mathcal{O}(|E|)$ time.*

**Proof**   For all $v \in V$, $Z_v$ is defined as the normalization if $v$ was the source in $\mathcal{G}$. Let $\mathcal{P}_v$ be the set of all $k$-multipaths sourced from $v$ and sinking at $\mathcal{T}$. Then:

$$Z_v = \sum_{\boldsymbol{\pi} \in \mathcal{P}_v} W(\boldsymbol{\pi}) \exp(-\eta \, \boldsymbol{\pi} \cdot \boldsymbol{\ell}).$$

For a sink node $v \in \mathcal{T}$, the normalization constant is vacuously 1 since no normalization is needed. For any non-sink $v \in V - \mathcal{T}$, we "peel off" the first multiedge leaving $v$ and then recurse:

$$Z_v = \sum_{\boldsymbol{\pi} \in \mathcal{P}_v} W(\boldsymbol{\pi}) \exp(-\eta \, \boldsymbol{\pi} \cdot \boldsymbol{\ell})$$
$$= \sum_{m \in M_v} \sum_{\substack{\boldsymbol{\pi} \in \mathcal{P}_v \\ \text{starts with } m}} W(\boldsymbol{\pi}) \exp(-\eta \, \boldsymbol{\pi} \cdot \boldsymbol{\ell}).$$

Now, we can factor out the weight and exponentiated loss associated with multiedge $m \in M_v$. Assume the $k$-multiedge $m$ comprised of $k$ edges from the node $v$ to the nodes $u_1, \ldots, u_k$. Notice,

excluding $m$ from the $k$-multipath, we are left with $k$ number of $k$-multipaths from the $u_i$'s:

$$Z_v = \sum_{m \in M_v} \sum_{\substack{\boldsymbol{\pi} \in \mathcal{P}_v \\ \text{starts with } m}} W(\boldsymbol{\pi}) \exp(-\eta \, \boldsymbol{\pi} \cdot \boldsymbol{\ell})$$

$$= \sum_{m \in M_v} \underbrace{(w_m)^{\pi_m} \exp(-\eta \, \pi_m \sum_{e \in m} \ell_e)}_{\widehat{w}_m} \sum_{\substack{(\boldsymbol{\pi}_1, \dots, \boldsymbol{\pi}_k) \in \\ \Pi_{u_1} \times \dots \times \Pi_{u_k}}} \prod_{i=1}^{k} W(\boldsymbol{\pi}_{u_i}) \exp(-\eta \, \boldsymbol{\pi}_{u_i} \cdot \boldsymbol{\ell}).$$

Observe that since the $\boldsymbol{\pi}_{u_i}$'s are independent for different $u_i$'s, we can turn the sum of products into product of sums:

$$Z_v = \sum_{m \in M_v} \widehat{w}_m \sum_{\substack{(\boldsymbol{\pi}_1, \dots, \boldsymbol{\pi}_k) \in \\ \Pi_{u_1} \times \dots \times \Pi_{u_k}}} \prod_{i=1}^{k} W(\boldsymbol{\pi}_{u_i}) \exp(-\eta \, \boldsymbol{\pi}_{u_i} \cdot \boldsymbol{\ell})$$

$$= \sum_{m \in M_v} \widehat{w}_m \prod_{i=1}^{k} \underbrace{\sum_{\boldsymbol{\pi} \in \Pi_{u_i}} W(\boldsymbol{\pi}) \exp(-\eta \, \boldsymbol{\pi}_u \cdot \boldsymbol{\ell})}_{Z_{u_i}}$$

$$= \sum_{m \in M_v} \widehat{w}_m \prod_{i=1}^{k} Z_{u_i}. \tag{1}$$

Now for each $v \in V - \mathcal{T}$, for all $m \in M_v$, set $w_m^{\text{new}} := \widehat{w}_m \frac{\prod_{u:(v,u) \in m} Z_u}{Z_v}$. The second property of Definition 3 is true since:

$$\sum_{m \in M_v} w_m^{\text{new}} = \sum_{m \in M_v} \widehat{w}_m \frac{\prod_{u:(v,u) \in m} Z_u}{Z_v}$$

$$= \frac{1}{Z_v} \sum_{m \in M_v} \widehat{w}_m \prod_{u:(v,u) \in m} Z_u$$

$$= \frac{1}{Z_v} \times Z_v = 1. \qquad \text{Because of Equation (1)}$$

We now prove that the first property of Definition 3 is also true:

$$\prod_{m \in M} (w_m^{\text{new}})^{\pi_m} = \prod_{v \in V - \mathcal{T}} \prod_{m \in M_v} (w_m^{\text{new}})^{\pi_m}$$

$$= \prod_{v \in V - \mathcal{T}} \prod_{m \in M_v} \left( \widehat{w}_m \frac{\prod_{u:(v,u) \in m} Z_u}{Z_v} \right)^{\pi_m}$$

$$= \left[ \prod_{v \in V - \mathcal{T}} \prod_{m \in M_v} (\widehat{w}_m)^{\pi_m} \right] \left[ \prod_{v \in V - \mathcal{T}} \prod_{m \in M_v} \left( \frac{\prod_{u:(v,u) \in m} Z_u}{Z_v} \right)^{\pi_m} \right].$$

Notice that $\prod_{v \in V - \mathcal{T}} \prod_{m \in M_v} \left( \frac{\prod_{u:(v,u) \in m} Z_u}{Z_v} \right)^{\pi_m}$ telescopes along the $k$-multiedges in the $k$-multipath. After telescoping, since $Z_v = 1$ for all $v \in \mathcal{T}$, the only remaining term will be $\frac{1}{Z_s}$

where $s$ is the souce node. Therefore we obtain:

$$\prod_{m \in M} (w_m^{\text{new}})^{\pi_m} = \left[ \prod_{v \in V - \mathcal{T}} \prod_{m \in M_v} (\widehat{w}_m)^{\pi_m} \right] \left[ \prod_{v \in V - \mathcal{T}} \prod_{m \in M_v} \left( \frac{\prod_{u:(v,u) \in m} Z_u}{Z_v} \right)^{\pi_m} \right]$$

$$= \left[ \prod_{m \in M} (\widehat{w}_m)^{\pi_m} \right] \left[ \frac{1}{Z_s} \right]$$

$$= \frac{1}{Z_s} \prod_{m \in M} (\widehat{w}_m)^{\pi_m} = W^{\text{new}}(\boldsymbol{\pi}).$$

Regarding the time complexity, we first focus on the the recurrence relation $Z_v = \sum_{m \in M_v} \widehat{w}_m \prod_{u:(v,u) \in m} Z_u$. Note that for each $v \in V$, $Z_v$ can be computed in linear time in terms of the number of outgoing edges from $v$. Thus the computation of all $Z_v$'s takes $\mathcal{O}(|E|)$ time. Now observe that $w_m^{\text{new}}$ for each multiedge $m \in M$ can be found in $\mathcal{O}(k)$ time using $w_m^{\text{new}} = \widehat{w}_m \frac{\prod_{u:(v,u) \in m} Z_u}{Z_v}$. Hence the computation of $w_m^{\text{new}}$ for all multiedges $m \in M$ takes $\mathcal{O}(|E|)$ time since $|M| \times k = |E|$. Therefore the generalized weight pushing algorithm runs in $\mathcal{O}(|E|)$.

$\square$

## C   Relative Entropy Projection to the $k$-Flow Polytope

Formally, the projection $\boldsymbol{w}$ of a given point $\widehat{\boldsymbol{w}} \in \mathbb{R}_{\geq 0}^{|E|}$ to constraint $C$ is the solution to the following:

$$\operatorname*{arg\,min}_{\boldsymbol{w} \in C} \sum_{e \in E} w_e \log \left( \frac{w_e}{\widehat{w}_e} \right) + \widehat{w}_e - w_e.$$

$C$ can be one of the three types of constraints mentioned in Definition 4. We use the method of Lagrange multipliers in all three cases. Observe that if $k = 1$, then the third constraint is non-existent and the updates in Koolen et. al. [25] are recovered.

### C.1   Constraint Type (i)

The outflow from the source $s$ must be $k$. Assume $w_{e_1}, \ldots, w_{e_d}$ are the weights associated with the outgoing edges from the root. Then:

$$L(\boldsymbol{w}, \lambda) := \sum_{e \in E} w_e \log \left( \frac{w_e}{\widehat{w}_e} \right) + \widehat{w}_e - w_e - \lambda \left( \sum_{j=1}^{d} w_{e_j} - k \right)$$

$$\frac{\partial L}{\partial w_e} = \log \frac{w_e}{\widehat{w}_e} = 0 \longrightarrow w_e = \widehat{w}_e \qquad \forall e \in E - \{e_1, \ldots, e_d\}$$

$$\frac{\partial L}{\partial w_{e_j}} = \log \frac{w_{e_j}}{\widehat{w}_{e_j}} - \lambda = 0 \longrightarrow w_{e_j} = \widehat{w}_{e_j} \exp(\lambda) \tag{2}$$

$$\frac{\partial L}{\partial \lambda} = \sum_{j=1}^{d} w_{e_j} - k = 0. \tag{3}$$

Combining equations (2) and (3) results in normalizing $w_{e_1}, \ldots, w_{e_d}$, that is:

$$\forall j \in \{1..d\} \quad w_{e_j} = \frac{k \, \widehat{w}_{e_j}}{\sum_{j'=1}^{d} \widehat{w}_{e_{j'}}}.$$

## C.2 Constraint Type (ii)

For a given multiedge $m \in M$, let $w_0^{(m)}, \ldots, w_{k-1}^{(m)}$ be the weights of the $k$ edges in $m$. Assuming $j^- = j - 1 \pmod k$ and $j^+ = j + 1 \pmod k$, then:

$$L(\boldsymbol{w}, \lambda) := \sum_{e \in E} w_e \log\left(\frac{w_e}{\widehat{w}_e}\right) + \widehat{w}_e - w_e - \sum_{m \in M} \sum_{j=0}^{k-1} \lambda_j^{(m)}(w_j^{(m)} - w_{j^-}^{(m)})$$

$$\frac{\partial L}{\partial w_j^{(m)}} = \log \frac{w_j^{(m)}}{\widehat{w}_j^{(m)}} - \lambda_j^{(m)} + \lambda_{j^+}^{(m)} = 0 \longrightarrow w_j^{(m)} = \widehat{w}_j^{(m)} \frac{e^{\lambda_j^{(m)}}}{e^{\lambda_{j^+}^{(m)}}} \qquad \forall j \in \{0, 1, \ldots, k-1\} \tag{4}$$

$$\frac{\partial L}{\partial \lambda_j^{(m)}} = w_j^{(m)} - w_{j^-}^{(m)} = 0 \longrightarrow w_j^{(m)} = w_{j^-}^{(m)} \qquad \forall j \in \{0, 1, \ldots, k-1\}. \tag{5}$$

Combining equations (4) and (5), for all $m \in M$ and for all $j \in m$, we can obtain:

$$\left(w_j^{(m)}\right)^k = \prod_{j'=0}^{k-1} w_{j'}^{(m)} = \prod_{j'=0}^{k-1} \widehat{w}_{j'}^{(m)} \longrightarrow w_j^{(m)} = \sqrt[k]{\prod_{j'=0}^{k-1} \widehat{w}_{j'}^{(m)}}.$$

which basically indicates that each weight must be assigned to the geometric average of the weights the edges in its multiedge.

## C.3 Constraint Type (iii)

Given any internal node (i.e. non-source and non-sink), the outflow from the node must be $k$ times of the inflow of that node. Assume $w_1^{(in)}, \ldots, w_a^{(in)}$ and $w_1^{(out)}, \ldots, w_b^{(out)}$ are the weights associated with the incoming and outgoing edges from/to the node $v$, respectively. Then:

$$L(\boldsymbol{w}, \lambda) := \sum_{e \in E} w \log\left(\frac{w_e}{\widehat{w}_e}\right) + \widehat{w}_e - w_e - \lambda\left(\sum_{b'=1}^{b} w_{b'}^{(out)} - k \sum_{a'=1}^{a} w_{a'}^{(in)}\right)$$

$$\frac{\partial L}{\partial w_e} = \log \frac{w_e}{\widehat{w}_e} = 0 \longrightarrow w_e = \widehat{w}_e \quad \forall e \text{ non-adjacent to } v$$

$$\frac{\partial L}{\partial w_{b'}^{(out)}} = \log \frac{w_{b'}^{(out)}}{\widehat{w}_{b'}^{(out)}} - \lambda = 0 \longrightarrow w_{b'}^{(out)} = \widehat{w}_{b'}^{(out)} \exp(\lambda) \qquad \forall b' \in \{1..b\} \tag{6}$$

$$\frac{\partial L}{\partial w_{a'}^{(in)}} = \log \frac{w_{a'}^{(in)}}{\widehat{w}_{a'}^{(in)}} + k\lambda = 0 \longrightarrow w_{a'}^{(in)} = \widehat{w}_{a'}^{(in)} \exp(-k\lambda) \qquad \forall a' \in \{1..a\} \tag{7}$$

$$\frac{\partial L}{\partial \lambda} = \sum_{b'=1}^{b} w_{b'}^{(out)} - k \sum_{a'=1}^{a} w_{a'}^{(in)} = 0. \tag{8}$$

Letting $\beta = \exp(\lambda)$, for all $a' \in \{1..a\}$ and all $b' \in \{1..b\}$, we can obtain the following by combining equations (6), (7) and (8):

$$\beta\left(\sum_{b'=1}^{b} \widehat{w}_{b'}^{(out)}\right) = \frac{k}{\beta^k}\left(\sum_{a'=1}^{a} \widehat{w}_{a'}^{(in)}\right) \longrightarrow \beta = \sqrt[k+1]{k \frac{\sum_{a'=1}^{a} \widehat{w}_{a'}^{(in)}}{\sum_{b'=1}^{b} \widehat{w}_{b'}^{(out)}}}$$

$$w_{b'}^{(out)} = \widehat{w}_{b'}^{(out)}\left(k \frac{\sum_{a''=1}^{a} \widehat{w}_{a''}^{(in)}}{\sum_{b''=1}^{b} \widehat{w}_{b''}^{(out)}}\right)^{\frac{1}{k+1}}, \qquad w_{a'}^{(in)} = \widehat{w}_{a'}^{(in)}\left(\frac{1}{k} \frac{\sum_{b''=1}^{b} \widehat{w}_{b''}^{(out)}}{\sum_{a''=1}^{a} \widehat{w}_{a''}^{(in)}}\right)^{\frac{k}{k+1}}.$$

This indicates that to enforce the $k$-flow property at each node, the weights must be multiplicatively scaled up/down so that the out and inflow will be proportionate to the $k$-to-1 weighted geometric average of the outflow and inflow, respectively. Concretely:

$$\text{outflow} := \sqrt[k+1]{k \, (\text{outflow})^k \, (\text{inflow})}, \quad \text{inflow} := \frac{1}{k} \sqrt[k+1]{k \, (\text{outflow})^k \, (\text{inflow})}.$$

# D   CH Regret Bound on $k$-Multipaths

**Proof**   According to Koolen, Warmuth and Kivinen [25], with proper tuning of the learning rate $\eta$, the regret bound of CH is:

$$\mathbb{E}[L_{\text{CH}}] - L^* \leq \sqrt{2\,L^*\,\Delta(\boldsymbol{\pi}||\boldsymbol{w}^{\text{init}})} + \Delta(\boldsymbol{\pi}||\boldsymbol{w}^{\text{init}}), \tag{9}$$

where $\boldsymbol{\pi} \in \mathbb{N}^{|E|}$ is the best $k$-multipath and $L^*$ its loss. Define $\widehat{\boldsymbol{w}}^{\text{init}} := \frac{1}{|V|^2}\,\mathbf{1}$ where $\mathbf{1} \in \mathbb{R}^{|E|}$ is a vector of all ones. Now let the initial point $\boldsymbol{w}^{\text{init}}$ be the relative entropy projection of $\widehat{\boldsymbol{w}}^{\text{init}}$ onto the $k$-flow prolytope[6]

$$\boldsymbol{w}^{\text{init}} = \arg\min_{\boldsymbol{w}\in P} \Delta(\boldsymbol{w}||\widehat{\boldsymbol{w}}^{\text{init}}).$$

Now we have:

$$
\begin{aligned}
\Delta(\boldsymbol{\pi}||\boldsymbol{w}^{\text{init}}) &\leq \Delta(\boldsymbol{\pi}||\widehat{\boldsymbol{w}}^{\text{init}}) && \text{Pythagorean Theorem}\\
&= \sum_{e\in E} \pi_e \log \frac{\pi_e}{\widehat{w}_i^{\text{init}}} + \widehat{w}_i^{\text{init}} - \pi_e \\
&= \sum_{e\in E} \pi_e \log \frac{1}{\widehat{w}_i^{\text{init}}} + \pi_e \log \pi_e + \widehat{w}_i^{\text{init}} - \pi_e \\
&\leq \sum_{e\in E} \pi_e (2\log|V|) + \sum_{e\in E} \pi_e \log \pi_e + \sum_{e\in E} \frac{1}{|V|^2} - \sum_{e\in E} \pi_e && (10)\\
&\leq D(2\log|V|) + D\log D + |E|\frac{1}{|V|^2} - \sum_{e\in E} \pi_e \\
&\leq 2D\log|V| + D\log D.
\end{aligned}
$$

Thus, by Inequality (9) the regret bound will be:

$$\mathbb{E}[L_{\text{CH}}] - L^* \leq D\sqrt{2\,T\,(2\log|V| + \log D)} + 2D\log|V| + D\log D.$$

Note that if $\boldsymbol{\pi}$ is a bit vector, then $\sum_{e\in E} \pi_e \log \pi_e = 0$, and consequently, the expression (10) can be bounded as follows:

$$
\begin{aligned}
\Delta(\boldsymbol{\pi}||\boldsymbol{w}^{\text{init}}) &\leq \sum_{e\in E} \pi_e(2\log|V|) + \sum_{e\in E} \pi_e \log \pi_e + \sum_{e\in E} \frac{1}{|V|^2} - \sum_{e\in E} \pi_e \\
&\leq D(2\log|V|) + |E|\frac{1}{|V|^2} - \sum_{e\in E} \pi_e \\
&\leq 2D\log|V|.
\end{aligned}
$$

Again, using Inequality (9), the regret bound will be:

$$\mathbb{E}[L_{\text{CH}}] - L^* \leq D\sqrt{4\,T\,\log|V|} + 2D\log|V|.$$

$\square$

# E   Additional Loss with Approximate Projection

First, let us define the notation $\preceq$. Given two vectors $\boldsymbol{a}$ and $\boldsymbol{b}$ of the same dimensionality, we say $\boldsymbol{a} \preceq \boldsymbol{b}$ iff $\boldsymbol{a}$ is less than $\boldsymbol{b}$ elementwise.

Now let us discuss approximate projection and additional loss. As we are working with inexact projection, we propose a slightly different prediction algorithm for CH. Suppose, using iterative Bregman projections, we reached at $\widehat{\boldsymbol{w}} \in \mathbb{R}_{\geq 0}^{|E|}$ which is $\epsilon$-close to the exact projection $\boldsymbol{w} \in \mathbb{R}_{\geq 0}^{|E|}$ in 1-norm, that is $\|\boldsymbol{w} - \widehat{\boldsymbol{w}}\|_1 < \epsilon$. Then do the following steps for prediction:

1. Set $\widetilde{w} := \widehat{w} + \epsilon \cdot \mathbf{1}$ where $\mathbf{1} \in \mathbb{R}_{\geq 0}^{|E|}$ is a vector of all ones.

2. Apply decomposition procedure on $\widetilde{w}$ and obtain a set of paths $\Pi$ and their associated coefficients $\{p_{\boldsymbol{\pi}} \mid \boldsymbol{\pi} \in \Pi\}$. Since $\widetilde{w}$ does not necessarily belong to the $k$-flow polytope, the decomposition $\Pi$ will not zero-out all the edges in $\widetilde{w}$:

$$\bar{w} := \sum_{\boldsymbol{\pi} \in \Pi} p_{\boldsymbol{\pi}} \cdot \boldsymbol{\pi} \preceq \widetilde{w}$$

3. Normalize and sample from decomposition $\Pi$.

First note that since $\|w - \widehat{w}\|_1 < \epsilon$ we have $|w_e - \widehat{w}_e| < \epsilon$ for all $e \in E$. Therefore $\widetilde{w} \succeq w$. This means that in the decomposition procedure, $w$ will be subtracted out from $\widetilde{w}$. Thus we have $\bar{w} \succeq w$ and

$$w \preceq \bar{w} \preceq \widetilde{w} = \widehat{w} + \epsilon \cdot \mathbf{1}. \tag{11}$$

Now let $z$ be the normalization constant for $\Pi$. Hence the expected prediction will be $\bar{w}/z$. Note that since $\bar{w} \succeq w$ and $w$ is in the $k$-flow polytope, then $z \geq 1$. Also notice that the weights of outgoing edges from the source $s$ in $\widetilde{w}$ are at most $2\,\epsilon$ greater than the ones in $w$ which belongs to the $k$-flow polytope. Thus the outflow at $s$ in $\widetilde{w}$ is at most $k + 2\,|V|\,\epsilon$. Therefore, since $\bar{w} \preceq \widetilde{w}$, we have $z \leq 1 + \frac{2|V|}{k}\,\epsilon$. Now we establish a closeness property between the approximate projected vector and the expected prediction vector with approximate projection:

$$\|\frac{\bar{w}}{z} - \widehat{w}\|_1 \leq \|\bar{w} - \widehat{w}\|_1 + \frac{z-1}{z}\|\bar{w}\|_1$$

$$\leq \|\bar{w} - \widehat{w}\|_1 + \frac{2|V|}{k}\,\epsilon\,\|\bar{w}\|_1.$$

Recall from Theorem 4 that $D$ is an upper-bound on the 1-norm of the $k$-multipaths. Using this upper bound, $\|\bar{w}\|_1$ can be bounded:

$$\|\bar{w}\|_1 \leq \|\widehat{w} + \mathbf{1} \cdot \epsilon\|_1 \leq \|w + \mathbf{1} \cdot 2\,\epsilon\|_1 \leq \|w\|_1 + 2|E|\epsilon \leq D + 2|E|\epsilon.$$

Therefore:

$$\|\frac{\bar{w}}{z} - \widehat{w}\|_1 \leq \|\bar{w} - \widehat{w}\|_1 + \frac{2|V|}{k}\,\epsilon\,\|\bar{w}\|_1$$

$$\leq \epsilon\,|E| + \frac{2|V|}{k}\,\epsilon\,(D + 2|E|\epsilon) = \epsilon\,(|E| + \frac{2|V|}{k}(D + 2|E|\epsilon)).$$

Next we establish closeness between the expected prediction vectors in exact and approximate projections:

$$\|\frac{\bar{w}}{z} - w\|_1 \leq \|\frac{\bar{w}}{z} - \widehat{w}\|_1 + \|\widehat{w} - w\|_1$$

$$\leq \epsilon\,(|E| + \frac{2|V|}{k}(D + 2|E|\epsilon)) + \epsilon = \epsilon\,(1 + |E| + \frac{2|V|}{k}(D + 2|E|\epsilon)).$$

Now we can compute the total expected loss over the trials $t = 1 \ldots T$ using approximate projection:

$$\left| \sum_{t=1}^{T} \frac{\bar{w}^{(t)}}{z} \cdot \ell^{(t)} \right| \leq \left| \sum_{t=1}^{T} w^{(t)} \cdot \ell^{(t)} \right| + \left| \sum_{t=1}^{T} (\frac{\bar{w}^{(t)}}{z} - w^{(t)}) \cdot \ell^{(t)} \right|$$

$$\leq \left| \sum_{t=1}^{T} w^{(t)} \cdot \ell^{(t)} \right| + \sum_{t=1}^{T} \left\| \frac{\bar{w}^{(t)}}{z} - w^{(t)} \right\|_1 \cdot \left\| \ell^{(t)} \right\|_{\infty}$$

$$\leq \left| \sum_{t=1}^{T} w^{(t)} \cdot \ell^{(t)} \right| + T \times \epsilon\,(1 + |E| + \frac{2|V|}{k}(D + 2|E|\epsilon)) \times 1.$$

For $\epsilon \leq \frac{1}{T\,(1 + |E| + \frac{2|V|}{k}(D + 2|E|\epsilon))}$ we have at most one unit of additional loss compared to the expected cumulative loss based on exact projections.

## Footnotes

[5]We are over-counting the number of cuttings. The number of possible cutting is called *partition function* which is approximately $e^{\pi\sqrt{2n/3}}/4n\sqrt{3}$ [10]. Thus if we run the Hedge algorithm inefficiently with one weight per cutting, we will get better regret bound by a factor of $\sqrt[4]{n}$.

[6]This computation can be done as a pre-processing step.