[Reviews · NeurIPS 2017]

Reviewer 1



The paper extends the celebrated multiplicative update algorithm to combinatorial structured problems whose offline versions can be solved some kind of DP algorithms. More precisely, the authors extend the Expanded Hedge and the Component Hedge algorithms to k-multipaths and applications thereof. The extension is well conducted by inspecting the various steps of these algorithms. The extension is made so that similar type of regret guarantees are achieved. The technical contribution of the paper lies in the simplification of the characterization of the convex hull of the decision space due to the underlying DP structure. Except for the introduction, the paper is well presented and easy to follow. The authors describe more and more involved models and algorithms, which help a lot understanding the contributions. The introduction focuses on BST, which is only one application of the proposed framework, and confusing actually … The main concerns about the paper is (i) the lack of reference to existing work on online Markov Decision Processes (arguably one of the main applications of the proposed framework), and (ii) the lack of numerical experiments. (i) For example, it seems that similar setting as the one described in Section 4 has been investigated in Even-Dar et al. in “Online markov decision processes”, Maths of OR, 2009. How do your results compare to those of the latter paper? There have been plenty of papers on this topic since 2009, and some of them with full information feedback as that considered here. Overall the paper is interesting and provides a nice way of generalizing Hedge algorithms to combinatorial problems whose offline version can be solved using DP. However, this does not seem to be a new topic, and the lack of comparison with existing work is regrettable.

Reviewer 2



The aim of this paper is to extend results in online combinatorial optimization to new combinatorial objects such as $k$-multipaths and binary search trees. Specifically, the authors are interested in extending Expanded-Hedge (EH) and Component Hedge (CH) to online combinatorial games, for which the offline problem can be solved in a bottom-up way using Dynamic Programming. Though the results about online optimization with $k$-multipaths are interesting, the overall paper could benefit from more polishing: some definitions and notations are not clearly defined, and some assertions look incorrect. Namely: In Section 4, the set (of subsets) $\mathcal H_v$ is not clearly defined, so it is very difficult for the reader to capture the family of DP tasks defined according to the recurrence relation (3). In this section, some concrete examples of online optimization problems characterized by this DP family would be helpful. In Section 5, the notion of Binary Search Tree should be clearly defined. Notably, if the authors are interested by the set of all rooted binary trees with $n + 1$ leaves, the cardinality of this set is indeed the Catalan number of order $n$ (as indicated in Line 268). But, contrary to what is claimed in Lines 250-251, there is a well-known characterization of the polytope specified by the convex hull of all rooted binary trees of order $n$: this is the $n$th dimensional associahedron. Since this polytope is integral, we can use (at least theoretically) the Component Hedge algorithm for projecting points onto this polytope, and decomposing them (by Caratheodory's theorem) into convex combinations of binary trees. Section 5.2 is of particular importance, as it explains how the online BST problem can be solved using previous results obtained for $k$-multipaths. But the translation of this problem into an online $2$-multipath learning task is not clear at all. To this very point, it is important to explain in unambiguous terms how we can learn BSTs using the CH (or EH) technique used for $k$-multipaths. According to the sets $V$, $\Omega$, and $F_{ij}$ defined in Lines 265-266, are we guaranteed that (i) every $k$-multipath of the induced graph encodes a rooted binary tree with $n+1$ leaves (correctness), and (ii) every possible rooted binary tree with $n+1$ leaves can be encoded by a $k$-multipath of this induced graph?

Reviewer 3



This work proposes an online dynamic programming method for a number of problems which involve an exponential number of combinatorial objects. The idea is to reduce the subproblems in dynamic programming algorithms to vertices in k-regular DAG, so that a concise representation of the combinatorial objects can be constructed. The paper is focused on the optimal binary search tree problem but the authors also show how other problems can be addressed by this new technique in the appendix. Overall, I think this is a solid work. A question is, can the authors explicitly discuss what kinds of problems on which this algorithm can be applied? Some notations are not well explained, e.g. some notations in Eq.(3). There are some language errors in the paper, such as “can also visited” in line 119 and a plural issue in line 211.